# A Comprehensive Review for the Surveillance of Human Pathogenic Microorganisms in Shellfish

**DOI:** 10.3390/microorganisms11092218

**Published:** 2023-08-31

**Authors:** Marion Desdouits, Yann Reynaud, Cécile Philippe, Françoise S. Le Guyader

**Affiliations:** Ifremer, Unité Microbiologie Aliment Santé et Environnement, RBE/LSEM, 44311 Nantes, France; marion.desdouits@ifremer.fr (M.D.); yann.reynaud@ifremer.fr (Y.R.); cecile.philippe@ifremer.fr (C.P.)

**Keywords:** seafood microbial contamination, pathogen emergence, surveillance, bivalves

## Abstract

Bivalve molluscan shellfish have been consumed for centuries. Being filter feeders, they may bioaccumulate some microorganisms present in coastal water, either naturally or through the discharge of human or animal sewage. Despite regulations set up to avoid microbiological contamination in shellfish, human outbreaks still occur. After providing an overview showing their implication in disease, this review aims to highlight the diversity of the bacteria or enteric viruses detected in shellfish species, including emerging pathogens. After a critical discussion of the available methods and their limitations, we address the interest of technological developments using genomics to anticipate the emergence of pathogens. In the coming years, further research needs to be performed and methods need to be developed in order to design the future of surveillance and to help risk assessment studies, with the ultimate objective of protecting consumers and enhancing the microbial safety of bivalve molluscan shellfish as a healthy food.

## 1. Introduction

Bivalve molluscan shellfish (BMS) filter large volume of waters for their physiological activities, such as feeding or respiratory. Their natural habitat is usually on coastal areas, where they find nutrients and other conditions needed for their development. Either considered as an easy food to collect that provide nutrients, or as an economic activity, they have been consumed for centuries and still constitute an important food source and commercial activity across the world [1,2]. On the other hand, coastal areas that have always been a place for human habitats will experience an increase in human density in the coming decades. As a correlate, the issue of coastal seawater contamination with both human sewage and animal sewage due to different activities, such as farming and agricultural practices, may increase. Food and water quality are considered important drivers that may favor the emergence of new pathogens and transmissions of infectious agents [3]. Close interactions between humans, animals and the environment were identified as an important factor that may increase the risk of pathogen emergence [4].

At present, oysters contaminated with norovirus are the seafood most often implicated in outbreaks [5], but do we know all the other microbial pathogens that can be detected in BMS? In this article, we did not set out to be exhaustive in regard to the huge amount of data describing the pathogenic microorganisms detected in BMS, but rather to highlight the diversity of the bacterial and viral pathogens identified in this food worldwide and the challenges ahead for an improved surveillance strategy. We also wanted to describe how new methods can help to describe the variety of microorganisms present in BMS, with the ultimate aim of improving the quality of these foods and thus preventing further epidemics or even the potential emergence of new pathogens.

## 2. Historical Considerations on Shellfish Contamination by Pathogenic Microorganisms

Since the prehistoric ages, mollusks have been considered a food, as evidenced, for example, by some work performed in the Aegean costal area [6]. More precisely, bivalve molluscan shellfish hold a critical and special role for poor communities living close to the sea who have collected this easy-to-catch food for centuries [7]. Many pieces of art and paintings confirm the important place of BMS in the history of many different countries. In France, evidence of oyster consumption has been established through oyster shell discoveries, some dated from the second century [8]. Several centuries later, in September 1716, king Louis XV signed an ‘ordonnance du Roy’ that forbade people to collect mussels, oysters and any other types of shellfish that grow on harbor breakwater, confirming the interest of the local population in these BMS. In the following years, several official publications began to control oyster distributions for consumption. In November 1731, the first report of an oyster-related outbreak in Paris led to the ban of shellfish commercial activity for a month and to the first set up of control on marketed oysters, albeit without any clear criteria as the microbiological impact on human health was not known [9].

By the end of the 19th century, evidence of the impact of oyster consumption on human health was raised in France, as well as in Italy, Great Britain and the USA (Connecticut). Most of the outbreaks were due to *Salmonella typhimurium* and the link between oysters harvested or stored in waters contaminated with human sewage began to be mentioned. Despite several observations in different countries, it was difficult to really understand when and why some oysters presented a risk for human consumption. In 1897, the French minister of Public Health asked to perform a study on the different production areas and to eliminate the ones for which human sewage contamination was clear. A few years later, the demonstration that purification for 10 to 20 days with seawater purified through a sand filter eliminates bacteria led the French Academy of Medicine to propose to purify all oysters put on the market [9]. 

In fact, it was a private association that began to set up microbiological analysis in 1913. Based on what was performed in New Zealand, a group of people (one with a seafood restaurant in Paris, la Maison Prunier) decided to conduct analysis to favor producers that market good quality oysters and to protect consumers. Two years later, a public law described some prescriptions to follow for shellfish farms and set up a network of laboratories to perform analysis [10]. Rapidly, the number of typhoid fever outbreaks linked to oyster consumption decreased; however, some outbreaks occurred due to the re-immersion of oysters in sewage-contaminated waters. Progressively, different regulations set in France and in the different countries producing BMS led to a clear decrease in outbreaks linked to bacteria, while viral gastroenteritis outbreaks persisted. Viral hepatitis outbreaks linked to oyster consumption began to be described in several countries, such as Sweden, USA, Italy, and India [11,12]. Gastroenteritis outbreaks were also described, but laboratory analysis failed to identify any potential agents inducing the symptoms. One of the first pieces of evidence of the implication of BMS in viral gastroenteritis cases was described in the UK, with cooked cockles linked to about 800 cases [13]. Since these first descriptions, evidence of BMS as a vector of disease in consumers has been described worldwide [5,14]. 

Indeed, BMS sold for human consumption still present some microbial contamination, despite regulations. For example, between January 2020 and January 2023 (included), a total of 207 alerts regarding microbial contamination in molluscan bivalve products were notified to the European portal [15]. Among these, 182 samples provided all of the information, such as country of production, microbial agent detected and date of report. Twelve samples were imported from non-European Union (EU) countries (Canada, Norway, UK, Vietnam, Turkey and South Korea) and notified for the detection of *Escherichia coli* contamination (2), norovirus (NoV) (4) and *Salmonella* (6). All of the other notifications concerned European products, such as mussels (32%), oysters (44.5%), burrowing shells (clams, cockles, tellinas) (22.5%) and a few others, such as whelk and razor shell (one sample for each). For the three complete years, 56 notifications were reported in 2020, 46 in 2021 and 49 in 2022. Regarding the season distribution, 31% were reported in winter (without the last 19 samples in 2023), 18% in spring, 20% in summer and 30% in autumn. *Vibrio parahaemolyticus* were detected twice in mussel samples (one in autumn and one in winter), hepatitis A virus once in a mussel sample in summer and rotavirus once in an oyster sample in autumn. All of the other notifications concerned *E. coli*, *Salmonella* and norovirus (Appendix A). Norovirus was mainly reported in the spring and winter seasons, and mainly in oyster samples. *E. coli* was more frequently reported in autumn, while *Salmonella* was less frequently detected, with no clear seasonal distribution, and mainly in burrowing shellfishes and mussel samples (only one positive sample was reported in oyster).

The FAO and WHO have recently published technical guidance for the development of Bivalve Mollusk Sanitation Programs regarding growing areas [16]. This document provides recommendations for microbiological monitoring programs including sanitary surveys, sampling plans, sampling and sample transport, laboratory testing, data handling and storage and the interpretation of data. Regarding microbiological testing, the European reference method ISO 16649-3 (Most Probable Number (MPN)) use *E. coli* as a fecal indicator. Alternative methods may be used, but must first be validated according to ISO 16140-2:2016. These alternative methods in the EU include the impedance approach and colony count methods (based on ISO 16649-2). Such monitoring systems of BMS production areas enable: i. the identification of unusual contamination by fecal bacteria; ii. the classification of harvesting areas, which determines whether the areas can be used for harvesting, and the level of post-harvest treatment; iii. the analysis of trends in contamination by fecal bacteria over time. These data are useful for assessing, for example, the impact of water treatment policies and infrastructures on BMS contamination and water quality. In addition, historical data can help to objectify site selection for research projects targeting the emergence of microbial pathogens for humans. Nevertheless, such programs monitoring a single fecal indicator are not capable of detecting and quantifying emerging microorganisms, not even norovirus, which is well-known for its involvement in foodborne illness outbreaks. In the general context of global warming and the increasing population density, particularly along the coast, we can expect an increase in anthropogenic discharges of potentially pathogenic viruses and bacteria into the marine environment. To anticipate such deterioration, there is an urgent need to develop tools targeting these microorganisms that are applicable to monitoring programs of BMS harvesting areas.

## 3. Shellfish May Be Contaminated by a Large Diversity of Pathogens

### 3.1. Marine and Enteric Bacteria

Among the bacterial pathogens, the *Vibrio* species, including *V. parahaemolyticus*, *V. cholerae* and *V. vulnificus*, have been extensively studied as key agents responsible for shellfishborne diseases, especially in shellfish like oysters, mussels and ark shells, and particularly during warmer months, when their populations surge [17,18,19]. More recent research has shed light on *Aeromonas* pathogenic species, with studies emcompassing the virulence and emergence of drug-resistant *Aeromonas* in seafood [20,21,22]. The authors advocate for paying increased attention to these pathogens, particularly considering the context of global warming and its effect on water temperatures.

Furthermore, Enterobacteriaceae regroups important seafood-borne pathogens, and include genera like *Escherichia* [23,24], *Shigella* which was reported in Japan [25], *Salmonella* [26,27,28,29,30,31] and *Klebsiella* in Portugal [32], Tunisia [33] and Italy [34], which have drawn attention to their health threats and their contribution to the spread of antibiotic resistance in the environment (Figure 1 and Appendix A).

*Salmonella* is a zoonotic pathogen transmitted from animals to humans through foodstuffs and can contaminate shellfish through fecal contamination from animals or contaminated water sources. Similarly, *Campylobacter* bacteria share the spotlight with Salmonella as major causes of foodborne illness and can also be found in shellfish, suggesting contamination from environmental sources like water and bird excreta. They encompass species such as *C. jejuni*, which is associated with poultry and bovines, playing a prominent role in human infections; *C. coli*, linked to poultry and pork; and *C. lari*, which is a more recent group [19,35,36,37,38]. Recent studies showed that *C. lari* is particularly associated with shellfish contamination [36,39]. The risk of infection from the consumption of raw shellfish was estimated to be 5–20% for mussels and 2–10% for oysters [35].

Recent reports have highlighted the presence of *Helicobacter pylori* in Spanish commercial shellfish [40] (Figure 1), adding to concerns about food safety. This bacterium, known for its role in causing gastric ulcers and other gastrointestinal disorders in humans, being detected in shellfish raises questions about the potential health risks associated with shellfish consumption. Additionally, pathogenic *Arcobacters* like *A. butzleri* and *A. cryaerophilus* have garnered attention for their association with seafood-related infections and their ability to develop antibiotic resistance [41,42]. *A. butzleri* has commonly been associated with various animals, including poultry, swine, cattle and shellfish (Figure 1), as well as with some human infections. On the other hand, *A. cryaerophilus* has been detected in a variety of sources, such as the environment, food products and animal intestines, but its association with human infections has been less well-established compared to *A. butzleri*. This may be due, in part, to their optimal growth temperature, as *A. butzleri* tends to grow optimally at temperatures around 30–37 °C, while *A. cryaerophilus* has an optimal growth temperature range of 25–30 °C [43,44].

In addition, gram-positive bacteria also inhabit shellfish. *Enterococcus* was found in cockles, oysters and mussels (Figure 1), and serves as a widely utilized indicator organism for assessing water contamination, particularly in recreational waters such as beaches and lakes [19]. Several species were reported in shellfish in Northern Ireland, with *Enterococcus faecalis* and *E. faecium* being the most frequently encountered [45] (Figure 1). Of note, four isolates of *E. faecalis* from the shellfish used in this study were highly resistant to vancomycin (Vancomycin-resistant enterococci, or VRE). Usually encountered in hospitals and long-term care facilities, this resistance poses a significant challenge as VRE infections are difficult to treat and can spread easily. 

Alongside *Enterococcus*, *Clostridium* bacteria have been identified in shellfish (Figure 1). While *C. butyricum* is generally considered to be non-pathogenic [46], *C. perfrigens* [47,48] and *C. difficile* [49,50,51,52] pose a significant risk of causing severe gastrointestinal infections. The anaerobic or microaerobic growth requirements of certain pathogens render their isolation for risk assessment challenging. 

Notably, a considerable number of the cited pathogens belong to the ESKAPEE group, which includes *E. faecium*, *Staphylococcus aureus*, *Klebsiella pneumoniae*, *Acinetobacter baumannii*, *Pseudomonas aeruginosa*, *Enterobacter* spp. and *E. coli*. These opportunistic pathogens are also considered to be emerging pathogens and serve as significant reservoirs of antibiotic resistance genes.

### 3.2. Human and Animal Enteric Viruses

The first attempts to detect human viruses in BMS relied on virus cultivation in mammalian cells, which allowed culturable viruses such as enteroviruses [53] or hepatitis A virus [54] to be identified. Since the 1990s, progress in molecular biology has enabled the identification of many other human viruses in BMS worldwide (Figure 2 and Appendix A) through conventional PCR, PCR and probe hybridization, quantitative PCR and, more recently, digital PCR. Most of these human viruses are considered to be enteric pathogens and known to be present in the stool of infected individuals and in human sewage [55]: adenovirus (AdV), aichivirus (AiV), astrovirus (AsV), enterovirus (EV), hepatitis A virus (HAV), hepatitis E virus (HEV), norovirus (NoV GI, GII, GIV), rotavirus A (RV) and sapovirus (SaV). Other non-enteric viruses that are present in human excreta were also detected in BMS, such as SARS-CoV-2 (Figure 2 and Appendix A), human polyomavirus JC [56,57] and the Merkel-cell polyomavirus [58]. In addition to these well-known human viruses, human bocaviruses, which are emerging enteric and/or respiratory viruses first identified in 2005 and are present in human sewage, were detected in BMS in all of the studies in which they were assessed [59,60,61,62,63], including in Italy, South-Africa, Thailand and Brazil. Human circoviruses, another group of recently discovered ubiquitous viruses also present in human stools, were detected in BMS in Norway [64].

Together, these studies report the worldwide exposure of BMS to human sewage, and show that BMS can be contaminated by multiple viral species, provided that these viruses are circulating in the local population and present in sewage. They also highlight the need for more data in poorly studied regions of the world, mainly Africa and South and Central America, but also North America and Oceania, where few studies have been conducted (or published) on viral contaminants in BMS and those few studies pertained to a limited set of human viruses. Finally, they show that emerging or newly discovered viruses (HEV, SARS-CoV-2, human bocavirus and circovirus) could be detected in BMS, which emphasizes the potential contribution of BMS in the propagation of emerging pathogens. 

Most of the human viruses assessed in BMS could be found, according to these studies (Appendix A), but some were detected more sporadically than others. For instance, human NoV were detected in most of the studies in which they were screened for (Figure 2, red full circles), while HEV and SARS-CoV-2 often remained undetected (Figure 2, green and gray empty circles). Interestingly, 12 studies reported the screening of multiple human viruses in the same BMS sample set, allowing their respective frequencies to be compared in identical settings (Figure 3) [56,60,65,66,67,68,69,70,71,72,73,74]. Human NoV (GI, GII or both) were detected in all of these studies but one and were the most or second most frequent virus in each (Figure 3). Human AdV and EV were always detected when assessed, albeit with varying frequencies. Conversely, human AiV, AsV, HAV, HEV, RV and SaV were detected in a smaller proportion of samples and remained undetected in some studies. Hence, human NoV appear as one of the most frequent human viruses in BMS, which is consistent with the vast amount of publications regarding this contamination, as reviewed in [14]. However, other human viruses are often present in NoV-contaminated shellfish, and sometimes in NoV-negative ones (Figure 3). Thus, the current focus on NoV in BMS likely masks the contribution of other viruses. The different frequencies of human viruses in BMS may reflect their respective levels of circulation in the human population, concentration in stool and sewage, persistence in the environment and/or affinity for shellfish tissues [75]. More studies are needed to assess these different hypotheses and to better understand the drivers of BMS contamination by the different human viruses present in sewage. In turn, this knowledge will help in designing adequate surveillance systems considering the viral threats that are relevant locally.

In addition to human viruses, BMS were found to be contaminated by mammalian animal viruses belonging to the same viral families—such as porcine or bovine NoV [76,77,78], porcine SaV [77], bovine polyomavirus [58], G8P [1] bovine-like rotavirus [79] or porcine circovirus [80]—in studies using targeted means, such as quantitative PCR. Metagenomics approaches to sequencing all of the viral material in a sample were also applied to BMS in one study where other animal viruses related to AsV or NoV were detected [81]. Targeted metagenomics applied to contaminated BMS revealed that a high diversity of NoV strains belonging to various genotypes can be present in a single sample, including strains of animal origin [82,83,84]. The same approach has shown the presence of animal, mainly avian, astroviruses in mussels [85]. This highlights the potential of BMS to contribute to the transmission of zoonotic viruses and the need to consider this risk for future surveillance strategies.

Detecting and identifying the multiple viruses and viral strains within a viral genus that can be present in a single BMS sample represents a challenge for the future surveillance of BMS quality, for which next-generation sequencing approaches hold much promise, as discussed below. Another important challenge is to evaluate the actual infectious risk posed by a contaminated BMS when some viral genomic material is detected.

### 3.3. Bacteria-Virus Interactions and Co-Occurrence

The studies reviewed above have shown the large diversity of bacterial and viral pathogens contaminating BMS worldwide. Meanwhile, the current monitoring strategies still focus on *E. coli* as a marker of fecal contamination. In some studies, there was a modest correlation between *E. coli* and NoV in BMS, especially in winter [86,87,88]. Indeed, the *E. coli*-based classification of production areas is able to partially protect against the commercialization of NoV-contaminated BMS [88] and, in Europe, BMS from class A areas were less frequently contaminated by NoV than those from class B areas [89]. However, several studies have shown that this marker is not well-correlated to contamination by human NoV in BMS [19,90,91]. This is likely due to differences in their dissemination in the land–sea continuum [92,93] and different interactions with the BMS matrix—*E. coli* contamination being transient, while viruses are more stable [94]. Moreover, *E. coli* is not always correlated to other enteric bacteria in BMS [19,95] and is unlikely to be correlated to marine bacteria like *Vibrio*; however, to the best of our knowledge, such a relationship has not been studied to date. Similar concerns have been raised in subtropical regions [96]. Beyond *E. coli* and NoV, few studies have addressed the possible correlation between human enteric bacteria and viruses in BMS, while they originate from the same contamination source—sewage. These first data suggest that interactions may be species- and season-specific [19,88]. This in agreement with reports of direct interactions between human NoV and some strains of enteric bacteria (reviewed in [97]), which may also exist for other viruses. Understanding the possible patterns and mechanisms of co-occurrence between the different bacterial and viral pathogens contaminating BMS remains a challenge, but could help in conceiving optimized surveillance strategies for multiple pathogens in BMS.

## 4. How to Translate Microbial Analysis in Term of Public Health Issue?

As mentioned above, several bacteria can be found in shellfish, but only a few will actually be pathogenic to humans due to virulence factors. In shellfish, the pathogenic *E. coli* strains that can be present are typically non-Shiga toxin-producing *E. coli* (non-STEC) or generic *E. coli* strains. While generic *E. coli* itself may not cause infections, its presence indicates the potential risk of other pathogenic microorganisms that could lead to gastrointestinal infections if the shellfish are consumed raw or undercooked. Nevertheless, a study reported that STEC and enteropathogenic *E. coli* (EPEC) were found in 5% (12/238) and 8% (19/238) of shellfish samples, respectively, in France [98]. EPEC were also detected in Brazil [99]. EPEC can be confirmed by presenting the *eaeA* gene encoding the Intimin adherence protein or *ehxA*, encoding enterohemolysin A, and STEC with the *stx* genes encoding the shiga toxin or *saa* encoding STEC autoagglutining adhesin [98]. As mentioned earlier, *E. coli* are part of ESKAPEE pathogens and can evolve antimicrobial resistance, leading to the emergence of highly virulent strains.

Among the emerging pathogens, antimicrobial resistance (AMR) poses a considerable threat. Different patterns of antibiotic resistance (e.g., *E. coli*, *Salmonella enterica*, *Vibrios*) have been observed [100]. Interestingly, extended-spectrum β-lactamase (ESBL) production has been reported in several bacteria found in shellfish [32]. ESBLs mainly belong to the *Enterobacteriaceae* family and possess an enzymatic mechanism that allows them to inactivate a broad range of beta-lactam antibiotics. This activity extends to various penicillins, cephalosporins and monobactams. ESBL bacteria are often associated with healthcare-associated infections, but they can also be released into the environment and easily spread through horizontal gene transfer. In particular, ESBL-Vibrios have been detected in shellfish [101]. In addition to AMR, to determine the pathogenic Vibrios, researchers typically focus on investigating their virulence factors. *V. cholerae*, for instance, produces cholera toxin (encoded by the *ctx* gene) [102]. *V. parahaemolyticus* produces various virulence factors, such as hemolysins and adhesion factors. Usually the thermostable direct hemolysin (*tdh*) and thermostable related hemolysin (*trh*) are used as PCR targets [102,103]. Lastly, *V. vulnificus* produces hemolysins (hemolysin A can by researched through targeting the *vvha* gene), proteases and cytotoxins, contributing to its pathogenic nature and ability to cause tissue destruction [104]. Additionally, colistin resistance has been identified in STEC and *Salmonella* isolated from shellfish [105]. Finally, isolates of *K. michiganensi* from clams were found to carry a *K. pneumoniae* carbapenemase (KPC) encoding gene, deriving from highly virulent *K. pneumoniae* usually found in hospital environments. KPCs refer to *K. pneumoniae*, which possess the enzymatic capability to hydrolyze and deactivate carbapenem antibiotics. The KPC enzyme is particularly worrisome as it can inactivate one of the last lines of defense against serious bacterial infections, leaving limited treatment options for affected individuals. KPCs represent a critical public health concern due to their significant resistance to multiple classes of antibiotics, including carbapenems, which are considered the most potent and broad-spectrum antibiotics available. This acquired resistance is usually conferred by the presence of mobile genetic elements, such as plasmids, that carry the KPC genes capable of disseminating through horizontal gene transfer. On the other hand, Vancomycin-Resistant Enterococci (VRE) were detected in oysters in Northern Ireland. They have developed resistance to the antibiotic vancomycin but the mechanism was not investigated; this could be due to producing an efflux pump or acquiring plasmid-mediated resistance genes [45].

To monitor this emerging threat linked to AMR, novel approaches like high throughput qPCR screening of known antimicrobial resistance genes are being developed, including the analysis of shellfish samples [106].

While pathogenic species, further serotyping and characterization can be conducted, virulence markers can also be researched; for example, in *Salmonella* isolates from shellfish through using in silico methods or PCR. One of the most critical virulence factors in *Salmonella* is the Type III Secretion System (T3SS), a specialized molecular syringe that injects bacterial effector proteins directly into host cells. These invasion proteins, such as invasins and intimin, manipulate the host cell’s signaling pathways, enabling the bacteria to enter and replicate within the host cells. Virulence genes targeted by PCR include *invA* [107], a component of the *Salmonella* pathogenicity island 1, which is crucial for efficient cell invasion [108,109]. *fimA* can also be researched, as it is necessary for the assembly of type 1 fimbriae [110], and the enterotoxin gene from *S. typhimurium* (*stn*) can also be targeted [111]. In *Shigella*, the virulence factors also mostly rely on the T3SS. Additionally, the Invasion Plasmid Antigens (Ipa proteins) are the primary effectors injected into host cells by the T3SS. They play a central role in promoting bacterial invasion and spreading within the host’s intestinal lining. Ipa encoding genes, such as *ipaBCD*, can be detected through PCR [112]. Of note, the most commonly isolated Campylobacters, belonging to the *C. lari* group, have poorly documented pathogenicity [39]. Investigating their pathogenicity poses challenges, and there is a need to expand their diversity to better understand the lari-group. This is not an easy task, despite progress in microaerobic cultures, due to low concentrations as they do not multiply on foodstuffs and can enter a viable non-cultivable state [113].

A common feature among *Salmonella*, *Shigella* and *Campylobacter* is biofilm formation. These bacteria can create biofilms on surfaces, which promotes their survival in various environments and enhances their resistance to environmental stresses. *Salmonella*, in particular, exhibits notable resistance to various stresses, such as acidity or osmotic pressure [114], and it is likely to withstand salinity without hindering biofilm formation [115]. These aspects are crucial to consider, along with the ability of these pathogens to persist in shellfish until consumption.

Evaluating the risk regarding viral contamination is also an important challenge. To be infectious, a virus needs to recognize and enter the cell in which it will replicate. Most of the viruses implicated in outbreaks following BMS consumption to date are non-enveloped viruses. These viruses are simple particles constituted of a proteinaceous capsid and a genome (often RNA) and are deemed very persistent in the environment. The current detection methods for viruses in BMS (and in most food matrices) rely on molecular assays targeting a small portion of the genome, mainly quantitative (RT)-PCR, as exemplified by the norm for NoV and HAV [116]. While these techniques have the great advantage of being sensitive enough to detect traces of contaminating viruses in the complex food matrices, their main limitation is that they solely reveal the presence of the viral genome. The actual infectious risk linked to this genomic detection remains difficult to address. Indeed, many studies have shown that viral infectious titers decline faster than the genomic concentration measured through PCR in water or in foods (reviewed in: [117,118]). Before the genomes become altered and/or fragmented enough for the PCR to not be able to amplify its short target (100–200 bp in most qPCR assays), significant damage to the viral particles may arise, which impairs their ability to establish an infection [119,120]. Thus, the ratio between the genomic load and the infectious load varies with the delay between the contamination event and the detection, as well as with environmental conditions that may accelerate or slow down the viral decay [117,118].

Historically, virus detection in foods has relied on the isolation and in vitro culture of the foodborne viruses on permissive cells. In some studies, infectious viruses were successfully cultured from naturally contaminated BMS [53,54,79,121,122], which highlights the high infectious risks posed by these foods. This direct measure of the infectivity in the food sample was only possible for those viruses that replicate easily, to high titers and with a low detection threshold. In recent years, cell culture systems have been published for most of the human enteric viruses known to contaminate BMS—even those that were deemed not cultivable for decades, like human NoV [123], human SaV [124,125] and HEV [126,127]. However, several limitations still preclude the routine use of these cell-culture assays for the detection or assessment of infectious human viruses in BMS, including: high detection thresholds not compatible with the low viral contamination; lack of methods that recover infectious viral particles with; slow replication kinetics leading to long time to result; high costs of reagents and heavy workload for some cell-culture systems; specificity to some viral strains or genotypes, field strains not able to be replicated in vitro. One can cite, as an example, the culture of human NoV in human intestinal enteroids—an important breakthrough that has allowed access to the infectivity of human norovirus strains from several genotypes [123]. This system is reproducible across laboratories worldwide, but the reagents are expensive, the handling of 3D-cultured enteroids is labor-intensive, the virus replication requires an inoculum of at least 10^4^–10^5^ genome copies per assay—higher than the viral load in naturally contaminated shellfish—and some strains do not replicate for unknown reasons [123,128]. Therefore, this system cannot yet be used for the routine detection of infectious norovirus in BMS. However, it can be adapted for research pertaining to the persistence of human NoV in the coastal environment and in BMS [129,130].

To overcome the need for an easy and sensitive in vitro replication assay, several molecular approaches were developed to evaluate the integrity of the viral particle as a proxy for infectivity. Capsid integrity assays, also called “viability PCR”, are designed to measure only the viral genomes enclosed in viral capsids, having retained their function of genome protection (reviewed in [131]). Another approach is to assess the integrity of the viral genome itself using long-range PCR to amplify only the full-length genome and avoiding the fragmented ones [132]. These approaches often yield intermediate measures between a classical q(RT-)PCR on bulk nucleic acids and a measure of the viral infectious titer using cell culture methods [131,133]. In BMS, viability PCR was applied on human AdV, NoV, EV and HAV [133,134,135,136,137], where it could partially avoid the detection of thermally inactivated or naturally contaminating viruses. The sample preparation, viability dye concentration and incubation time need to be adapted to each virus and matrix. Of note, the current norm for NoV and HAV extraction from BMS includes a proteinase K digestion and a heating step, leading to viral inactivation [138,139]. Thus, to assess an infectious virus or capsid integrity, the extraction method needs to be adapted [140].

Another approach to evaluating the infectious potential of viruses in naturally contaminated BMS is to measure the infectious titer of a reporter virus present in the sample together with the target human virus. Such a virus should be: (i) easily cultivable; (ii) similar to the target enteric viruses in terms of its size and physico-chemical properties; (iii) decaying at the same rate, or slower, to avoid underestimating the infectious risk [141]. F-specific RNA bacteriophages were proposed as a reporter virus for infectious NoV in naturally contaminated BMS, where they behaved similarly at the genomic level [142,143]. Comparisons with infectious NoV or other naturally occurring human viruses are still needed to validate this approach.

The main challenge for the direct measure of infectious viruses in BMS remains the low level of the contamination. To overcome this, in research settings, the artificial contamination of BMS allows viral titers high enough to measure a viral decay to be reached. Following this approach, it was shown that infectious poliovirus titers decrease faster than those of HAV in BMS in a 23 h time-frame [144]. This confirms that the different viruses exhibit different behaviors in this matrix and highlights the need to obtain data for each viral contaminant. Surrogate viruses can be used instead of the target virus to model its persistence following artificial contamination. As an example, the Tulane virus was used as a surrogate for human NoV in BMS, where it remained infectious for up to 3 weeks [145]. However, in seawater, this surrogate tended to decay faster than infectious human NoV assessed using enteroids [129]. As was the case with the reporter viruses, the data obtained with surrogate viruses should be validated using the target human viruses when possible.

To summarize, the different approaches used to assess or model infectious human viruses all confirm that these viruses can remain infectious for extended periods of time in BMS, but also that they behave differently. Recent progress in cell culture systems has paved the way for obtaining crucial data with the actual target viruses, like human NoV, at least in research settings. Then, to translate the infectious titer into an infectious risk for consumers, one must consider the infectious dose of the virus, which is considered quite low for enteric viruses like NoV or HAV. Volunteer studies allowing this important parameter to be measured are scarce and difficult to set up, for obvious ethical and logistical reasons [146]. Importantly, epidemiological data obtained from actual outbreaks can be used in combination, which highlights the need to continue reporting on BMS-associated viral outbreaks, keeping in mind that low concentrations may induce a delay in symptoms and thus complicating the source identification [147,148]. Finally, regarding animal viruses, assessing their zoonotic potential is necessary as most will never cross species. For instance, in vitro studies performed after the detection of bovine NoV in oysters confirmed that the possible transmission to humans was unlikely due to the lack of a shared ligand between host species [78].

## 5. The Future of Surveillance: Anticipating Emergences?

A major issue in detecting any type of pathogens, including emerging ones, is the sampling strategy. Indeed, in an open environment such as coastal areas, the temporal and spatial variability of contamination may be important. This is considered by monitoring surveillance networks, which use a time series of data rather than a single analysis for the classification of production areas [149]. This is especially important for some bacteria that only stay in BMS for a few hours, while for human enteric viruses, which persist for a longer time, this may be less important. But many factors have to be considered, such as sedimentation, dilution, weather or currents [92]. Water circulation varies from one catchment to another and physical models are important to understanding the distribution of pathogens in order to optimize the sampling plan [150,151]. In such open environments, it is thus important to consider all of the different sources of contamination (sewage treatment plant outfall, broken pipes, animal waste, combined sewer overflows during weather events). The second most important point to consider is the BMS itself, as these live animals may present interspecies and interindividual variations in their filtering activity. Variability among animals has been reported for bacteria and for virus contamination [149,152]. Such variability may also be observed when performing replicate analysis within a batch, with no clear evidence of whether this can be linked to individual BMS variability or to a method issue, especially with low-contaminated samples [149].

### 5.1. Main Methodologies for Detecting and Characterizing Pathogenic Microorganisms in Seafood

#### 5.1.1. Targeted Strategies

Specific media for cultures of enterobacteria and other bacterial pathogens

The gold standard procedures still rely heavily on cultural methods and the purification of bacterial isolates before conducting serotyping, PCR and other characterizations. Over the years, a variety of tailored media have been developed for the cultivation of enterobacteria and various bacterial pathogens, owing to significant advancements in clinical microbiology, leading to more precise identification, antimicrobial susceptibility testing and epidemiological investigations. Among the selective and chromogenic media commonly utilized, such as MacConkey agar, Salmonella-Shigella (SS) agar and Xylose Lysine Desoxycholate (XLD), they present different interests to isolate bacteria based on their metabolic capacities and enable the differentiation of colonies’ morphotypes through color or growth capacity indications [153].

In recent years, advances in *Campylobacter* cultivation have facilitated their isolation on selective media and proper incubation conditions in a microaerophilic atmosphere, with reduced oxygen and increased carbon dioxide levels. Modified Charcoal Cefoperazone Deoxycholate Agar (mCCDA) facilitates the isolation of *Campylobacter* from food and environmental samples. Skirrow Agar is also used for the selective isolation of *C. jejuni* and *C. coli* [154].

Nevertheless, cultural methods demonstrate limitations due to the specific growth requirements of fastidious micro-aerophilic or anaerobic enteric bacteria and the presence of viable but non-culturable cells [113]. Molecular approaches offer valuable tools for addressing these challenges.

Screening by PCR technologies

While conventional colony counting enables specific bacterial species to be isolated, quantified and stored for subsequent studies, such as WGS (see below), these techniques, in their traditional format, are labor-intensive and only cultivable microorganisms can be isolated; therefore, *in fine*, this approach remains low-throughput.

PCR approaches, on the other hand, are fast, practical, sensitive and range from the detection to the absolute quantification of microorganisms, but do not provide the full genomic information of the strain of interest. At present, there is a whole range of PCR-based techniques, each with its own advantages and disadvantages. For example, multiplex PCR assays can simultaneously target two or more bacteria in a single reaction mixture, increasing the efficiency of pathogen detection compared to conventional PCR by reducing the detection time and costs. Li et al. [155], for example, recently designed a multiplex PCR system with good specificity and sensitivity for the detection of a total of eight food-borne bacterial pathogen species, including *V. parahaemolyticus*, *Listeria monocytogenes*, *Cronobacter sakazakii*, *Shigella flexneri*, *Pseudomonas putida*, *E. coli*, *V. vulnificus* and *V. alginolyticus*, targeting the *toxS*, *virR*, *recN*, *ipaH, rfbE, vvhA* and *gyrB1* genes, respectively. Nevertheless, this multiplex PCR approach does not quantify targeted amplicons, whereas quantitative PCR (qPCR) is able to extrapolate quantification using the Ct values and calibration curves. Several experiments using qPCR enable large spatial and temporal detection of pathogenic microorganisms from shellfish-harvesting areas in order to assess their prevalence and diversity. For example, Rincé et al. [19], during a two-year survey in France (English channel coast), followed the distribution of *E. coli* (phylogenetic groups A, B1, B2, or D), *enterococci*, *Salmonella*, *Campylobacter*, *Vibrio* (*V. cholerae*, *V. vulnificus* and *V. parahaemolyticus*) and HuNoV thanks to the application of several qPCR. As mentioned above, q(RT)-PCR is now the reference method for the detection of human viruses in shellfish [116]. In addition, quantitative PCR can be multiplexed for the simultaneous quantification of multiple pathogens, such as human and mammalian enteric viruses in BMS [76].

At present, high-throughput qPCR (HT-qPCR) using microfluidic technology has the capacity to process dozens of samples against dozens of targets; for example, a total of 9216 individual PCR reactions are performed in a single run using Standard BioTools’ Biomark and EP1 systems. This means that pathogenic species, virulent lineages, antibiotic resistance genes, specific plasmids, etc., can be screened in a single PCR run. This approach has been applied in many fields, for example, to detect SARS-CoV-2 variants and other pathogenic viruses in wastewater [156]; however, to the best of our knowledge, this HT-qPCR approach has not yet been developed for bacterial and viral screening in seafood. Such a development would be very useful for researchers working on health monitoring in the land-sea continuum.

In the field of quantitative PCR, digital PCR (dPCR) looks promising. This approach enables theoretically absolute quantification based on limiting dilution and Poisson distribution [157]. Compared with traditional qPCR, dPCR does not require a standard curve and reaches greater precision, reproducibility and sensitivity, particularly for rare target molecules, low copy numbers and rare variations. In addition, dPCR is less sensitive to inhibitors [158]. In BMS, this technique has been successfully developed to detect human NoV [159] and SaV [160]; however, to the best of our knowledge, it remains to be implemented for enteric bacteria. The absolute quantification and differentiation of pathogenic and non-pathogenic strains of *V. parahaemolyticus* in seafood, for example, are possible using a 4-plex dPCR to detect the tlh, tdh, ureR and orf8 genes in a single bacterial cell [161].

Characterization of micro-organisms by genomic approaches

Once cultivated, microbial strains can be fully sequenced through Whole Genome Sequencing (WGS). The democratization of sequencing technologies, both from an economic point of view and in terms of bioinformatics analysis and the availability of open-source software, make it possible to characterize circulating strains in depth, as well as the mechanisms underlying the emergence and persistence of potentially pathogenic strains. Applications in the field of genomics are numerous and include: the study of genetic variability, epidemiology and phylogenomics, phylogeography, exploration of core and accessory genomes, the identification of mobile genetic elements (plasmids, transposons, bacteriophages, etc.), genetic recombination, genetic support of virulence, resistance to antibiotics and heavy metals (resistome), evolutionary history (particularly using Bayesian models), niche adaptation, genetic selection, interactions between micro-organisms, source tracing, etc. While a plethora of studies have deciphered complex mechanisms and interactions in clinical fields worldwide, further research is needed to better explore the conditions that favor the circulation and emergence of human pathogenic microorganisms in the land–sea continuum, especially in seafood products. Nevertheless, in recent years, a growing number of studies have used genomics to address such issues in different bacterial models, such as *E. coli* [162], *Salmonella* [105,163,164], *K. pneumoniae* [165], *C. lari* [39], etc. For viruses, WGS approaches are less straightforward as isolation through in vitro culture is not currently frequently conducted. In BMS, targeted sequencing is mostly performed on short PCR products, allowing phylogenetic analyses and the characterization of viruses up to the genotype [70,166,167], but lacks the multiple applications brought by WGS.

#### 5.1.2. Non-Targeted Strategies

The disadvantage of traditional culture combined with WGS is that it only provides a restricted view of the microbial community structure. Instead, non-targeted approaches aim to explore the overall microbial diversity in a sample, from simple identification to the full characterization of genomes.

Metabarcoding

Metabarcoding targeting 16S rRNA genes has been used extensively to detect putative pathogens in the environment. Leight et al. [168], for example, studied the co-occurrence of fecal indicator bacteria with potential pathogenic bacterial genera in shellfish-growing areas of the Chesapeake Bay, USA, revealing the effect of rainfall on the microbial community composition in aid of *Enterobacteriaceae*, *Aeromonas*, *Arcobacter*, *Staphylococcus* and *Bacteroides*. Metabarcoding is nevertheless not quantitative, and the resolution can, at best, reach the genus level using technology like Illumina. Applying long read sequencing (Oxford Nanopore Technology (ONT) and PacBio) to metabarcoding analyses provides a solution as the long read lengths achieve the species-level taxonomic identification of pathogens, which previous short-read technologies could not accomplish. Using the MinION device, it is now possible to fully sequence 16S rRNA [169,170,171] and the 16S-ITS-23S operon [170,172], as well as to process a total of 96 samples in a single run.

16S rRNA metabarcoding data can be used to develop microbial source tracking (MST). These methods enable the identification of fecal pollution sources like human fecal material and contamination from birds or livestock [173]. Using Bayesian modeling like SourceTracker, it is also possible to estimate the source proportions and uncertainties on known and unknown sources [174].

For viruses, which lack a universal gene, several approaches have been designed using degenerate primers amplifying a conserved but variable portion of the viral genome as a barcode across a chosen range of viral families. The RNA-dependent RNA-polymerase (RdRp) is often used in environmental virology as it allows the fingerprinting of most families of RNA viruses, but it may lack the necessary resolution to identify specific genotypes inside a given viral genus. Thus, in BMS, metabarcoding-like approaches have targeted human viral species using primers to amplify portions of the capsid and/or RdRp genes, which could identify multiple genotypes of NoV [82,83,84] and AsV [85] inside a single sample.

Shotgun metagenomic

Metagenomics present great potential for both taxonomic and functional annotation, combining high-throughput and non-targeted capacities together with the deep exploration of near-whole genomes (metagenome-assembled genomes—MAG). An array of applications emerge from metagenomics [175]: temporal and spatial changes of microbiota [176], the surveillance of microbial pathogens [177] and public health risk assessment [178], the characterization of antimicrobial and heavy metals resistances genes [179,180], etc. Previous MAG studies of environmental [181,182] and gut microbiomes [183] revealed genomic particularities of uncultured bacterial lineages, like reduced genomes missing common biological functions (for example maintenance of osmotic pressure and protection against oxidative stress), slow replication and the absence of conserved genes (for example genes involved in fatty acid pathways). Such approaches could help in deciphering limitations in culture conditions and for the identification of novel growth factors for uncultured bacterial species, which is useful for the classical strategy combining culture and WGS, as well as for culturomic approaches. Another application concerns microbial network and transkingdom analyses, which have great potential for exploring interactions between microorganisms, like interactions between viral and bacterial communities. Several tools are available that enable the integration of different omics data (e.g., metabarcoding vs metagenomic data) in a single analysis [184]. Finally, the application of Bayesian source tracking has been successfully applied to metagenomic data from the coastal marine environment thanks to mSourceTracker [185]; this tool provides distinct source origins for distinct taxonomic groups and further determines the source proportions.

Most metagenomics studies use short-read sequencing technologies, but with the rise of third-generation long-read technologies applied to metagenomics, significant improvements in MAG reconstruction have begun [186].

When exploring microbial metagenomes in host samples, a significant challenge concerns the vast host DNA material that can dominate samples, in which, in turn, greatly reduce our capacity to recover the microbiome. The treatment of samples can be performed before metagenomic sequencing in order to increase the end microbial-to-host DNA ratio. For bacteria, these strategies are based on intrinsic differences between the host and microbial cells, like their sizes, wall structure and DNA methylation, and the use of filtration, centrifugation or chemical host depletions. Several host genome depletion kits or chemistries exist, as well as treatments with propidium monoazide (PMA). Some kits are designed for samples containing methylated host DNA, which are not supposed to be used for invertebrate genome depletion due to sparse methylation with a mosaic pattern in marine invertebrates genomes [187,188], while others perform the differential lysis of host vs bacterial cells [189,190]. Some have been successfully applied for bivalve symbiont enrichment [191]. Finally, osmotic lysis combined with PMA (lyPMA) seems to be a promising and cost-effective approach [192]. LyPMA has the potential to outperform enzymatic host genome depletion approaches, but this technique would nevertheless need to be tailored for each novel matrix under study.

For viruses, sample preparation includes a filtration step to remove the host and microbial cell; other steps, like sonication and free nucleic acid digestion, can also be added. The development of methods to deplete the host nucleic acids will be useful for virome analysis in BMS. Indeed, most viruses that contaminate BMS present short RNA genomes that are difficult to detect in such a complex matrix, especially regarding the low contamination.

Another way to significantly improve the abundance and diversity of microbial DNA is to perform targeted metagenomics using last-generation in-solution capture platforms. Such systems have successfully been applied in resistome studies to select antibiotics and heavy metals genes [193]. The same philosophy can be applied in viral metagenomics, where such capture systems often constitute a sine qua non condition to recover viruses present at very low concentrations in BMS samples [81,84,194]; it could also be developed in bacterial metagenomic studies. It should be kept in mind that these approaches cannot be qualified as «non-targeted».

Single cell microfluidic metagenomic

Finally, a new era is rising with the combination of the high throughput of metabarcoding/metagenomic with the high resolution of WGS: single cell omics could constitute the new paradigm shift in the microbial research field. For example, in 2022, Zheng et al. [195] proposed the so-called Microbe-seq, a high-throughput single-cell sequencing method that uses microfluidic to encapsulate single bacteria into droplets before microbial lyses, genome amplification and barcoding; the authors were able to study tens of thousands of microbes individually, i.e., with strain resolution and to fully reconstruct the associated genomes. Applying such a methodology to environmental samples would be of great interest.

Microbiome research and the combination of omics analyses will be useful in the coming years for:the systematic screening of putative emerging pathogens in the land–sea continuum;deciphering the mechanisms underpinning the circulation, dynamic, success and persistence/resilience of those pathogens;the detection of novel and powerful indicators of human contamination within coastal waters and seafoods.

## 6. Conclusions/Perspectives

In conclusion, we have shown that a large diversity of bacteria and viruses that are potentially pathogenic to humans have been detected in BMS worldwide. Progress in the next-generation sequencing approaches is likely to further unravel this diversity; however, many challenges remain for improving their sensitivity and representativity. In addition, the detection of microbial genomes overlooks the question of the actual infectious risk for consumers. Here, again, more data are needed to estimate the persistence of infectious pathogens or the virulence of some bacterial species. The main challenge ahead for the surveillance of BMS microbial quality lies in the frequent emergence of new human pathogens, which requires the constant optimization of detection methods and research on the behavior of these new pathogens in the land–sea continuum and in BMS. Indeed, even with the depuration process of shellfish, which is professionally handled, can we confidently consider the procedure as completely safe concerning all pathogenic species? Some viruses or bacteria can persist even after immersion in clean sea water [5], suggesting that further investigations are required to address the emerging and highly resistant pathogens and their persistence in bivalves. Another issue that was not detailed in this review is the association of human pathogens among one another. As mentioned earlier, only a few reports have aimed to investigate the presence of different bacterial species or interactions between bacteria and viruses. Additionally, a new chapter needs to be considered, focusing on the other pollutants, such as microplastics, that contribute to microbial aggregation, which are transported overseas and may facilitate the global sharing of pathogens [196].

Research development should be performed to design the future of surveillance and risk assessments and to help to optimize national or international policy, with the final objectives of protecting consumers and increasing the microbial of BMS, which should be a healthy food.

## Figures and Tables

**Figure 1 microorganisms-11-02218-f001:**
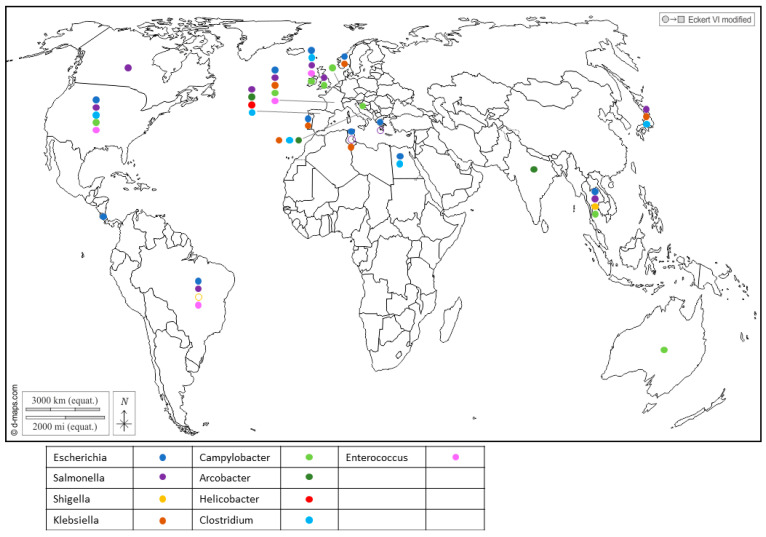
Worldwide distribution of bacterial genera frequently detected in BMS (freshwater or marine) per country. Full circles show the detection of bacteria in a country (by isolation or PCR), according to the color code in the legend (blue: *Escherichia*; purple: *Salmonella*; yellow: *Shigella*; brown: *Klebsiella*; light green: *Campylobacter*; dark green: *Arcobacter*; red: *Helicobacter*; light blue: *Clostridium*; pink: *Enterococcus*). Empty circles report that the bacterium was assessed but never detected in the country, following the same color code. Up to 5 different genera were detected in a single country. References reporting each detection are listed per country in Appendix A.

**Figure 2 microorganisms-11-02218-f002:**
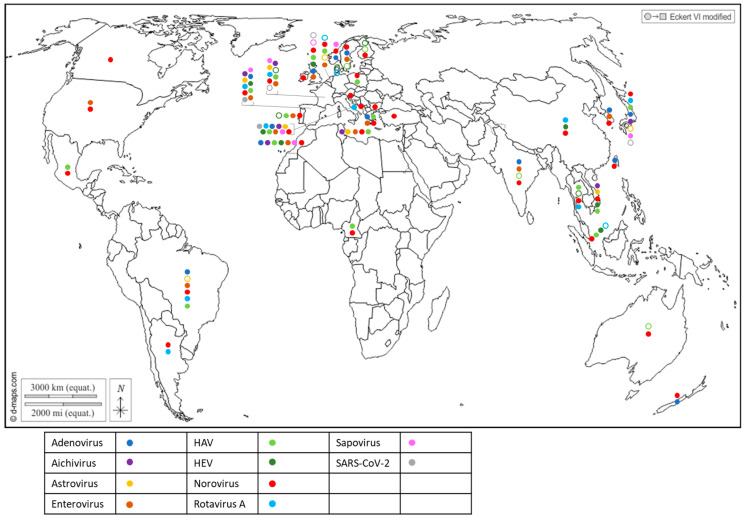
Worldwide distribution of 10 human viruses frequently detected in BMS (freshwater or marine) per country. Full circles denote the detection of a virus in a country, according to the color code in the legend (blue: Adenovirus; purple: Aichivirus; yellow: Astrovirus; brown: Enterovirus; light green: HAV; dark green: HEV; red: Norovirus; light blue: Rotavirus; pink: Sapovirus; grey: SARS-CoV-2). Empty circles report that the virus was assessed but never detected in the country, following the same color code. Up to 10 of these viruses were detected in a single country. References reporting each detection are listed per country in Appendix A.

**Figure 3 microorganisms-11-02218-f003:**
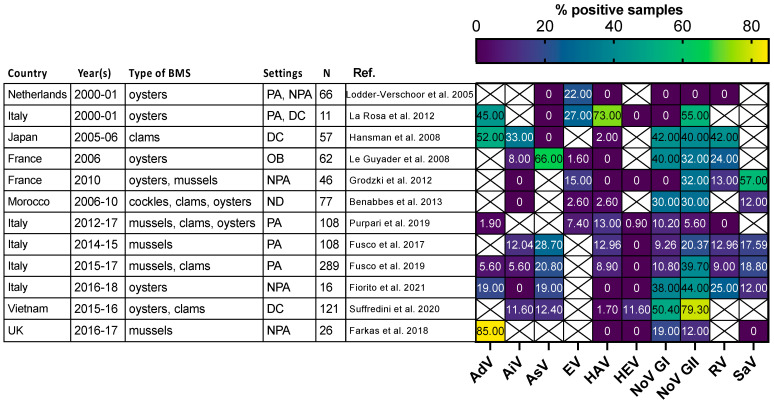
Comparison of enteric virus frequency in BMS screened for multiple viruses in 11 studies. The proportion of samples positive for each studied virus (*x*-axis) in 11 studies (*y*-axis) is plotted as a heatmap. Studies were selected for assessing at least 5 different enteric viruses through conventional or quantitative PCR. They were conducted in the Netherlands, Japan, France, Morocco, Italy, Vietnam and UK at different periods between 2000 and 2018. Different types of BMS (Oysters, mussels, clams and cockles) were sampled either from areas not open for production (NPA), producing areas (PA), the distribution chain (DC), which includes dispatch centers, retails, markets or restaurants, or in relation to human outbreak investigation (OB). In one case the setting was not disclosed (ND). The number of studied samples varied from 11 to 289. Human AdV and EV were consistently detected, with varying frequencies. Human NoV (GI, GII or both) were detected in all studies but one and often displayed the highest or second highest frequency. Other viruses were detected at lowest frequencies and remained undetected in some of them. References cited [56,60,65,66,67,68,69,70,71,72,73,74].

## Data Availability

Not applicable.

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
