# Peer review of "A Comprehensive Review for the Surveillance of Human Pathogenic Microorganisms in Shellfish"

_microorganisms, 2023, doi:10.3390/microorganisms11092218_

Round 1

Reviewer 1 Report

Dear Authors,

This manuscript presents a review related to the monitoring and control of human pathogenic microorganisms present in bivalves.

The manuscript is well written, comprising the abstract, keywords, six sections, three figures and references. Additionally, it includes two tables plus respective references, as supplementary material. Nevertheless, the utilization of ellipsis (...) may suggest an incomplete thought to the reader, so please avoid its use (lines 23, 25 & 497). Please avoid using contractions like “didn’t” (line 37). Please harmonize terms, such as “microorganims” (eg., lines 41, 508) vs. “micro-organims” (eg., lines 125, 130). Please avoid using abbreviations, if not used twice (eg., ARGs & VNC & VBNC, in lines 201 & 385 & 529).

The titles of some sections are quite long and have different punctuation, so it is advisable to make them shorter and harmonized, in order to give a more accurate and clear message to readers (eg., “1. Historical considerations on shellfish contamination by pathogenic microorganisms:”, “2. Shellfish may be contaminated by a large diversity of pathogens.”, “3. How to translate microbial analysis in term of public health issue?”). Also, please renumber sections, as there are two number 1.

The reference list is extensive (197) and valuable and corresponds with the citations. Please uniformize in-text citation format, as some are in bold (eg., lines 204-313, 400-486) whereas other are italicized (eg., lines 494-507).

Please see below some more specific comments.

Keywords

Perhaps could also include “bivalves”, in order to more accurately reflect the content.

1. Introduction

As “omics approaches”, previously referred in the abstract, can involve technologies that aim to comprehensively characterize/quantify large numbers of several biological molecules from microorganisms (also including: metabolomics, proteomics, transcriptomics, etc) in samples, please explain briefly in the introduction section, which omics will be addressed in the review, in order to make the text more clear to readers. 

Line  26  Please replace “they have... and still constitutes” by “they have... and still constitute”.

Line 34 – Please clarify what you mean by “ increase the risk of emergence. Perhaps you meant “increase the risk of pathogen emergence”?

1. Historical considerations on shellfish contamination by pathogenic microorganisms:

Lines 113-117  From the text, it isn’t clear that  ISO standard 16649-3 is the reference method for European Union Member States, but alternative methods can also be used if validated according to ISO standard 16140-2:2016. Besides France, for example the Netherlands [S. Rubini et al., 2023, Food Control 154(6):110005] is also using an alternative method (TBX pour plate method according to ISO standard 16649-2), which was validated against the reference method [I. Pol-Hofstad, W. Jacobs-Reitsma. Validation of the TBX pour plate method (ISO 16649-2) for the enumeration of Escherichia coli in Live Bivalve Molluscs: Renewal study for alignment with EN ISO 16140-2:2016. Rijksinstituut voor Volksgezondheid en Milieu RIVM (2021), 10.21945/RIVM-2021-0127]. So, please clarify the idea, give more EU examples and support with references.

2. Shellfish may be contaminated by a large diversity of pathogens.

Line 165 - Please replace “figure 1” by “Figure 1.

Line 213 - What do you mean by “Figure x and supplemental Table x”? Is it 2?

Line 298 - Please italicize the first “E. coli”.

4. How to anticipate emergence of pathogens in the land-sea continuum: the future of surveillance?

Line 502: Please add a space between “animals” and may”.

References

Please verify if all references follow the journal “Instructions for Authors”, in particular in what concerns “Books/Book Chapters” and revise accordingly (eg., reference 16).

Best Regards,

Reviewer

Author Response

Dear Authors,

This manuscript presents a review related to the monitoring and control of human pathogenic microorganisms present in bivalves.

The manuscript is well written, comprising the abstract, keywords, six sections, three figures and references. Additionally, it includes two tables plus respective references, as supplementary material.

Reply :We thank reviewer 1 for the positive feedbacks on the review and all the suggestions and corrections that aimed to improve clarity of the text for the readers.

Nevertheless, the utilization of ellipsis (...) may suggest an incomplete thought to the reader, so please avoid its use (lines 23, 25 & 497).

Reply: Modified accordingly (deleted as not really needed).

 Please avoid using contractions like “didn’t” (line 37).

Reply: Modified accordingly.

Please harmonize terms, such as “microorganims” (eg., lines 41, 508) vs. “micro-organims” (eg., lines 125, 130).

Reply: Modified accordingly.

Please avoid using abbreviations, if not used twice (eg., ARGs & VNC & VBNC, in lines 201 & 385 & 529).

Reply: Modified accordingly.

The titles of some sections are quite long and have different punctuation, so it is advisable to make them shorter and harmonized, in order to give a more accurate and clear message to readers (eg., “1. Historical considerations on shellfish contamination by pathogenic microorganisms:”, “2. Shellfish may be contaminated by a large diversity of pathogens.”, “3. How to translate microbial analysis in term of public health issue?”). Also, please renumber sections, as there are two number 1.

Reply: The section 4 title has been shortened. Sections have been renumbered.

The reference list is extensive (197) and valuable and corresponds with the citations. Please uniformize in-text citation format, as some are in bold (eg., lines 204-313, 400-486) whereas other are italicized (eg., lines 494-507).

Reply: Citation format was uniformized as suggested.

Please see below some more specific comments.

 Keywords

Perhaps could also include “bivalves”, in order to more accurately reflect the content.

Reply: We thank the reviewer for this suggestion, we included the keyword.

  1. Introduction

As “omics approaches”, previously referred in the abstract, can involve technologies that aim to comprehensively characterize/quantify large numbers of several biological molecules from microorganisms (also including: metabolomics, proteomics, transcriptomics, etc) in samples, please explain briefly in the introduction section, which omics will be addressed in the review, in order to make the text more clear to readers. 

Reply: We have replaced « omics approaches » by « genomics » to clarify.

Line  26 – Please replace “they have... and still constitutes” by “they have... and still constitute”.

Reply: Corrected accordingly.

 Line 34 – Please clarify what you mean by “ increase the risk of emergence”. Perhaps you meant “increase the risk of pathogen emergence”?

Reply: It is what we meant, we corrected accordingly.

  1. Historical considerations on shellfish contamination by pathogenic microorganisms:

Lines 113-117 – From the text, it isn’t clear that  ISO standard 16649-3 is the reference method for European Union Member States, but alternative methods can also be used if validated according to ISO standard 16140-2:2016. Besides France, for example the Netherlands [S. Rubini et al., 2023, Food Control 154(6):110005] is also using an alternative method (TBX pour plate method according to ISO standard 16649-2), which was validated against the reference method [I. Pol-Hofstad, W. Jacobs-Reitsma. Validation of the TBX pour plate method (ISO 16649-2) for the enumeration of Escherichia coli in Live Bivalve Molluscs: Renewal study for alignment with EN ISO 16140-2:2016. Rijksinstituut voor Volksgezondheid en Milieu RIVM (2021), 10.21945/RIVM-2021-0127]. So, please clarify the idea, give more EU examples and support with references.

Reply: Modified accordingly

  1. Shellfish may be contaminated by a large diversity of pathogens.

Line 165 - Please replace “figure 1” by “Figure 1.

Reply: Corrected accordingly

Line 213 - What do you mean by “Figure x and supplemental Table x”? Is it 2?

Reply: Corrected accordingly

Line 298 - Please italicize the first “E. coli”.

Reply: Corrected accordingly

  1. How to anticipate emergence of pathogens in the land-sea continuum: the future of surveillance?

Line 502: Please add a space between “animals” and “may”.

Reply: Corrected accordingly

References

Please verify if all references follow the journal “Instructions for Authors”, in particular in what concerns “Books/Book Chapters” and revise accordingly (eg., reference 16).

Reply: Books references were revised according to the journal’s guidelines.

Reviewer 2 Report

The review paper "A comprehensive review for the surveillance of human pathogenic microorganisms in shellfish" is a well-written paper that provides an overview of the most essential knowledge in the field of research on various pathogens that can be found in shellfish and that can have a harmful effect on humans. In addition to it provided an overview of the current techniques, this review warns about the shortcomings of the current methods of monitoring the health of shellfish and gives recommendations for the implementation of new strategies in order to achieve a higher level of health safety of shellfish for human consumption. Therefore, I believe this review paper is welcome, and I recommend its publication in the journal "Microorganisms".

However, I submit a few minor remarks that I ask the authors to consider:
Lines 23, 25, 497: I think using ellipses (...) in this kind of overview paper is inappropriate. Authors should state what they consider significant and not leave the reader to guess what the authors wanted to say.
Lines 35 – 36: Cite references that support these claims
Line 143: Usually, the  name of bacteria Family is not italicized
Line 213: Correct Table x and Image x
Line 502: ... live animals may ...

Author Response

Reviewer 2

The review paper "A comprehensive review for the surveillance of human pathogenic microorganisms in shellfish" is a well-written paper that provides an overview of the most essential knowledge in the field of research on various pathogens that can be found in shellfish and that can have a harmful effect on humans. In addition to it provided an overview of the current techniques, this review warns about the shortcomings of the current methods of monitoring the health of shellfish and gives recommendations for the implementation of new strategies in order to achieve a higher level of health safety of shellfish for human consumption. Therefore, I believe this review paper is welcome, and I recommend its publication in the journal "Microorganisms".

Reply: We thank reviewer 2 for the compliments on the manuscript and for all the comments that improved its overall quality.

However, I submit a few minor remarks that I ask the authors to consider:
Lines 23, 25, 497: I think using ellipses (...) in this kind of overview paper is inappropriate. Authors should state what they consider significant and not leave the reader to guess what the authors wanted to say.

Reply: Modified accordingly, we agree with this remark (also reported by reviewer 1).

Lines 35 – 36: Cite references that support these claims

Reply: The former ref. 13, now n°5 (Rowan et al, STOTEN, 2023) is now cited here.

Line 143: Usually, the  name of bacteria Family is not italicized

Reply: Corrected accordingly

Line 213: Correct Table x and Image x

Reply: Corrected accordingly

Line 502: ... live animals may ...

Reply: Corrected accordingly
